# Adaptive Tool Generation with Models as Tools and Reinforcement Learning

## Abstract

Tool-augmented language models have demonstrated strong capabilities, but their reliance on live API access creates scalability and reliability challenges during training and deployment. We propose MTR, a simulation-first training framework for tool-augmented reasoning. Instead of relying on live APIs, MTR learns from complete ReAct traces with schema-validated, simulated observations. Our approach operates through a multi-agent architecture where a ToolMaker generates task-specific, OpenAI-compatible tool interfaces, an AutoAgent produces structured think–act–observe sequences, and a ToolActor simulates realistic responses. Training proceeds in two stages: Stage-1 Supervised Fine-Tuning (SFT) teaches "trace grammar" from complete reasoning sequences; Stage-2 Group Relative Policy Optimization (GRPO) optimizes strategy with a composite trace reward that balances answer correctness and internal consistency. Across four multi-hop QA benchmarks (HotpotQA, MuSiQue, 2WikiMultiHopQA, Bamboogle), MTR attains competitive Exact Match (EM) scores to live-API systems and excels on reasoning-intensive tasks, suggesting that effective tool reasoning can be learned from structured traces without live interactions.

## 1 Introduction

Large language models (LLMs) have demonstrated remarkable capabilities across various reasoning domains, including mathematical, code, and logical reasoning (Ji et al., 2025; Shi et al., 2025; DeepSeek-AI, 2025). Nowadays, there has been growing interest in advancing LLM performance on multi-turn scenarios that require long-horizon planning and adaptive decision making skills over dynamic interactions with external tool environments. Among them, end-to-end Agentic Reinforcement Learning (Agentic RL), has emerged as a promising training paradigm for enhancing tool invoke, planning, and reasoning capabilities across diverse domains, including web search (Sun et al., 2025), deep research (Zheng et al., 2025), and general-purpose applications (Bai et al., 2025).

However, training LLMs for agentic reasoning faces two critical challenges that substantially impede progress. **Firstly**, the integration of external tools introduces significant training inefficiencies and computational overhead in multi-turn decision-making tasks (Asano et al., 2024; Sun et al., 2025). For example, in web search scenarios, agents rely on commercial search engines that incur monetary costs and execution latency, which severely constrains training throughput. Moreover, domain-specific applications require the development of specialized tools such as code execution environments, web browsers, and database query interfaces, further complicating implementation and increasing development costs. **Secondly**, agentic RL suffers from training instability and gradient explosion issues (Xue et al., 2025). Xue et al. (2025) observed that the incorporation of external tool feedback as model input in multi-turn interactions, causes significant distribution shift from the model's pretrained data distribution leading to the emergence and accumulation of extremely low-probability tokens.

To address these challenges, various approaches have been proposed from different perspectives. Some researchers focus on enhancing agent environment stability and scalability through synthetic or simulated tools (Zheng et al., 2024; Qiu et al., 2024; Liu et al., 2025; He et al., 2024). For example, Fan et al. (2025) proposes self-search RL within fully simulated search environments, which not only improves the internal knowledge utilization of LLMs but also enhances training efficiency. However, it only considers a single search scenario, thereby restricting its generalizability across diverse task domains. Regarding training stability, some efforts have focused on designing sophisticated opti-

mization algorithms (Dong et al., 2024) or directly filtering out failed trajectories (Xue et al., 2025), but neither approach fundamentally addresses the distributional shift introduced by external tool feedback. In this work, we propose **Model-as-Tools Reasoning (MTR)**, a novel multi-agent framework that employs models to simulate tools, thereby extending the scope of environment simulation in agentic RL. Specifically, our MTR incorporates three specialist agents: ToolMaker, ToolActor, and AutoAgent. The ToolMaker leverages task-specific information to construct specialized tools, thereby enhancing tool diversity and task completion accuracy. Subsequently, the ToolActor simulates tool execution processes and outcomes based on tools synthesized by the ToolMaker, dramatically improving training efficiency. Furthermore, we present a two-stage agent training framework. We first leverage our synthetic environment to efficiently generate large-scale trajectories for supervised fine-tuning (SFT) cold start. Subsequently, we employ end-to-end RL algorithms to further enhance the reasoning ability of the agent under our MTR framework. Since our tools are entirely model-simulated, we eliminate the distributional drift issues caused by external tool outputs, thereby achieving significantly more stable training dynamics. Empirical evaluation across four established multi-hop reasoning benchmarks validates the effectiveness of our trace-based learning paradigm. MTR achieves competitive average performance (29.38% vs. 29.3%) with live-API systems while demonstrating substantial improvements on reasoning-intensive evaluation, notably 40.0% exact match on Bamboogle compared to 33.3% for the strongest baseline.

To summarize, our key contributions are as follows:(1) We propose Model-as-Tool Reasoning (MTR), a novel framework that simulates tool definition and invocation for agentic RL, enabling scalable environment interaction without external API dependencies. (2) We introduce a two-stage training paradigm that strategically decouples structural learning from strategic optimization through composite reward mechanisms, enhancing both learning efficiency and performance stability. (3) We provide comprehensive empirical analysis demonstrating that our approach achieves superior performance while addressing scalability bottlenecks inherent in API-dependent approaches.

## 2 RELATED WORK

**Tool-augmented language models.** The integration of external tools has fundamentally expanded LLM capabilities beyond parametric knowledge. Early foundational work established key paradigms: MRKL formalized modular routing to experts and APIs (Karpas et al., 2022), Toolformer demonstrated self-supervised API integration (Schick et al., 2023), and ReAct popularized the *think–act–observe* loop that underpins many agent systems (Yao et al., 2023b). Large-scale systems like HuggingGPT and Gorilla enabled access to thousands of APIs but exposed brittleness from schema diversity and endpoint instability (Shen et al., 2023; Patil et al., 2023). Parallel approaches offload computation to program executors (PAL) or structured reasoning (Program-of-Thoughts) (Gao et al., 2022; Chen et al., 2022). Recent efforts focus on improving function calling through large-scale datasets (ToolBench, ToolAlpaca) and feedback-driven training (TRICE, ToolAlign) (Qin et al., 2023; Tang et al., 2023; Qiao et al., 2024; Chen et al., 2024). A common thread across these works is the reliance on live, runnable APIs during training and inference. Our work departs from this assumption: we show that effective tool-use policies can be learned from complete ReAct traces with simulated observations, eliminating API dependencies entirely.

**Learning from traces and reinforcement learning for reasoning.** Learning from structured traces has proven effective for complex reasoning tasks. Chain-of-thought and self-consistency approaches leverage step-by-step reasoning (Wei et al., 2022; Wang et al., 2022), while methods like Reflexion and Tree-of-Thoughts incorporate self-correction and search (Shinn et al., 2023; Yao et al., 2023a). Process rewards and step-by-step verification provide denser supervision than outcome-only signals. For optimization, recent advances in reinforcement learning have shown strong results on multi-step reasoning: Group Relative Policy Optimization (GRPO), popularized by DeepSeek-R1, uses group-weighted likelihood optimization with Kullback-Leibler (KL) regularization (DeepSeek-AI, 2025), while preference-based methods like Direct Preference Optimization (DPO) align outputs with human preferences (Rafailov et al., 2023). Simulation-based approaches have been explored in web and embodied environments (WebArena, ALFWorld) using specialized simulators for behavior cloning and offline RL (Zhou et al., 2023; Shridhar et al., 2021). We combine these insights in a two-stage approach: SFT on complete ReAct traces to learn "trace grammar," followed by GRPO with trace-level rewards targeting answer consistency and efficiency.

**Multi-hop reasoning and retrieval-augmented generation.** Multi-hop question answering requires compositional reasoning across multiple information sources. Datasets like HotpotQA, MuSiQue, 2WikiMultiHopQA, and Bamboogle evaluate multi-step synthesis and complex reasoning chains (Yang et al., 2018; Trivedi et al., 2022; Ho et al., 2020; Press et al., 2023). Retrieval-Augmented Generation (RAG) approaches improve factuality through external retrieval (DPR, FiD) and remain competitive on these benchmarks (Karpukhin et al., 2020; Izacard & Grave, 2021), though they typically optimize retrieval rather than tool orchestration. Recent tool-augmented systems achieve strong performance but require live API access to search engines and knowledge bases during both training and inference. Our results demonstrate that learning to generate ReAct traces with simulated tool responses can match or exceed live-API systems on these challenging tasks, suggesting that much of the benefit of tool augmentation can be captured through trace-based learning without external dependencies.

## 3 METHOD

The central challenge in tool-augmented reasoning lies in learning effective policies for tool selection and usage while avoiding the brittleness and computational overhead of live API dependencies. Our approach, Model-as-Tools Reasoning (MTR), builds on the key insight that complete reasoning traces encode both the structural patterns of tool interaction and the strategic decisions that underlie effective tool use. By systematically decoupling these two dimensions of learning, we achieve robust tool reasoning through a simulation-first paradigm that eliminates external dependencies.

MTR operates through a principled two-stage training methodology: (1) **Structural Competence Learning** via supervised fine-tuning (SFT) teaches models to generate syntactically valid and semantically coherent ReAct sequences, and (2) **Strategic Competence Optimization** via Group Relative Policy Optimization (GRPO) refines decision-making policies for tool selection, invocation timing, and error recovery patterns. This decomposition enables targeted optimization of each competency with appropriate supervision signals while preserving end-to-end differentiability.

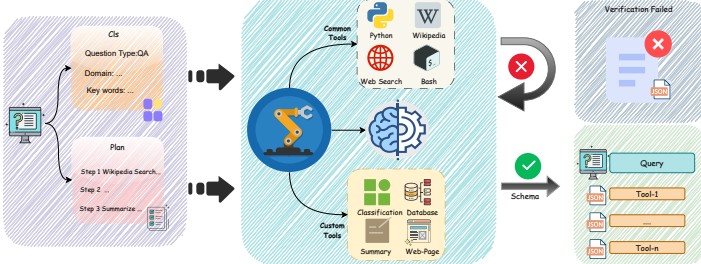

(a) ToolMaker workflow and tool generation process

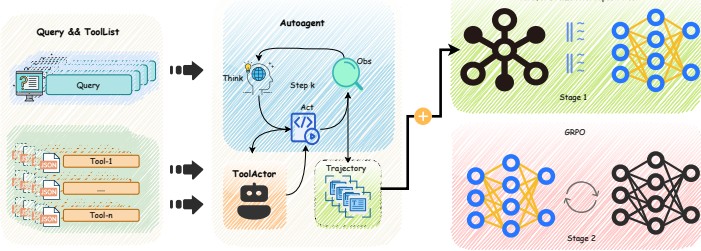

(b) Two-stage training pipeline for tool reasoning

Figure 1: **MTR Framework Components.** (a) The ToolMaker operates through a systematic pipeline that combines predefined utility tools with task-specific tool generation, including task classification, interface generation, and validation checking. (b) Our training methodology transforms ToolMaker-generated interfaces into complete reasoning policies through systematic competence separation using SFT for structural competence and GRPO for strategic competence.

## 3.1 MODEL-AS-TOOLS REASONING FRAMEWORK

Our framework produces high-quality reasoning traces through a coordinated multi-agent architecture. The key design principle follows the notion of *separation of concerns*: each agent is specialized for a distinct aspect of trace generation, thereby enabling scalable and controllable supervision signal creation. The MTR framework consists of three specialized agents that work together to create complete reasoning sequences without external API dependencies.

**Tool Interfaces and Validation.** We formalize tools as lightweight **interfaces** $\mathcal{I} = \{\mathcal{S}_{\text{in}}, \mathcal{S}_{\text{out}}, \mathcal{V}\}$, where $\mathcal{S}_{\text{in}}$ and $\mathcal{S}_{\text{out}}$ are JavaScript Object Notation (JSON) schemas for input and output (following OpenAI function calling format), and $\mathcal{V}$ is a set of lightweight validation checks (type checks, regex patterns, range constraints). This formalization serves two primary purposes: (1) it provides a structured interface for consistent tool interaction, and (2) it enables automatic validation that ensures well-formed traces during both inference and training.Formally, for a tool call $c$ with arguments $\mathbf{a}$, validation succeeds if $\mathcal{V}(\mathbf{a}, \mathcal{S}_{\text{in}}) = \text{True}$. If validation fails, a structured error response is generated, which subsequently triggers self-correction in the following reasoning steps. This mechanism enforces that all generated traces adhere to consistent and parseable patterns.

**Multi-Agent Trace Generation Protocol.** Trace generation is realized through three specialized agents that collectively construct complete and realistic ReAct sequences. Each agent is instantiated through carefully engineered prompts designed to enforce domain-specific expertise and maintain cross-agent consistency (see Appendices A,, and B Cfor detailed specifications):

1. **ToolMaker** ($\mathcal{M}$): Given a query $q$, this agent generates a set of tool interfaces $\mathcal{T} = \{\mathcal{I}_1, \ldots, \mathcal{I}_k\}$ tailored to the target task domain. Each interface follows OpenAI-compatible function calling format, with explicitly typed schemas and lightweight validation validation mechanisms. The agent follows professional tool design principles, ensuring coherence and appropriate granularity for the reasoning task. A representative example of this process is provided in Appendix D.

2. **AutoAgent** ($\mathcal{A}$): This agent is responsible for executing the core reasoning process by generating a structured sequence of think–act–observe steps. Conditioned on a query $q$ and the tool set $\mathcal{T}$, it produces actions $a_t \in \mathcal{T}$ and reasoning steps $r_t$, following systematic planning methodologies. An illustrative end-to-end reasoning trace exemplifying this process is presented in Appendix F.

3. **ToolActor** ($\mathcal{E}$): For each valid tool call $c_t$, this agent generates realistic observations $o_t$ that conform to the prescribed output schema $\mathcal{S}_{\text{out}}$. It operates exclusively from parametric knowledge, without access to ground-truth answers or evaluation data, thereby ensuring that the supervision signals are derived from realistic tool interactions rather than oracle information. Post-hoc verification is subsequently employed to evaluate trace quality based on final answer correctness.

**Empirical Validation of Tool Generation Intelligence.** Beyond standard performance metrics, we conduct a comprehensive evaluation of the generated tool ecosystem to assess the effectiveness of the ToolMaker agent. Our analysis examines tool usage patterns and generation coherence, demonstrating that the framework generates tools through systematic understanding rather than random sampling. These findings provide empirical support for the validity of our trace-based learning paradigm (see Appendix E for more details).

## 3.2 TWO-STAGE TRAINING FOR TOOL REASONING

**Problem Formulation.** We formulate tool reasoning as learning a policy $\pi_\theta$ that generates structured traces $\tau = (s_1, a_1, o_1, \ldots, s_T, a_T, o_T)$, where $s_t$ represents reasoning state, $a_t$ is a tool action (or reasoning step), and $o_t$ is the observed outcome. The challenge is learning this policy without access to live tool executions during training. MTR addresses this challenge through our simulation-first approach, where models act as both tool definers and tool executors.

Our training methodology is designed to address two fundamental questions: (1) *How can models learn the structural patterns of tool interaction?* and (2) *How can models develop strategic competence for tool selection and usage?* We argue that these objectiveness necessitate distinct forms

of supervision and therefore propose a two-stage training paradigm that explicitly separates these concerns.

### 3.2.1 Stage 1: Structural Competence via Supervised Fine-Tuning

**Objective.** The first stage aims to train the model to produce ReAct traces that are both syntactically well-formed and semantically coherent. Given a dataset of verified high-quality traces $\mathcal{D}_{\text{SFT}} = \{(q_i, \tau_i)\}$, where each trace $\tau_i$ is guaranteed to yield a correct answer (as determined through post-hoc verification), we optimize the standard maximum likelihood objective:

$$\mathcal{L}_{\text{SFT}}(\theta) = \mathbb{E}_{(q,\tau)\sim\mathcal{D}_{\text{SFT}}}\left[-\sum_{t=1}^{T}\log\pi_\theta(y_t|q, y_{<t})\right] \tag{1}$$

where $\tau = (y_1, \ldots, y_T)$ represents the tokenized trace sequence. This stage yields a reference policy $\pi_{\text{ref}}$ that can generate well-formed tool calls, parse responses correctly, and maintain coherent reasoning chains.

**Trace Quality Control.** To ensure reliable supervision, we filter generated traces based on three criteria: (1) correctness of the final answer after normalization using an independent verifier, (2) absence of validation errors, and (3) adherence to a reasonable trace length (between 2-12 tool calls). This retrospective filtering process retains approximately 60% of the generated traces, ensuring clean supervision signals while preserving the simulation-first paradigm where tool responses are generated without access to oracle access.

### 3.2.2 Stage 2: Strategic Competence via Group-Policy Optimization

**Objective.** The second stage optimizes the reasoning strategy, including decisions regarding tool selection, invocation timing, and error recovery. Building upon the SFT checkpoint, we apply Group-Policy Optimization (GRPO) (DeepSeek-AI, 2025), a reinforcement learning method that has demonstrated strong effectiveness in reasoning tasks involving long-from generation.

For each query $q$, we sample $n = 8$ candidate traces $G = \{\tau_1, \ldots, \tau_n\}$ from the policy $\pi_\theta$. The policy is updated to maximize reward-weighted likelihood subject to KL regularization:

$$\mathcal{J}(\theta) = \mathbb{E}_{G\sim\pi_\theta}\left[\sum_{i=1}^{n}w(\tau_i)\log\pi_\theta(\tau_i|q)\right] - \beta\cdot\mathbb{E}_q[D_{\text{KL}}(\pi_\theta(\cdot|q)||\pi_{\text{ref}}(\cdot|q))] \tag{2}$$

**Composite Trace Reward Design.** The reward function incorporates an answer extraction mechanism that evaluates both intermediate reasoning steps and final answers. For each trace $\tau$, answers are extracted from two sources: (1) the final answer enclosed within structured `<answer>` tags, and (2) intermediate predictions derived from reasoning steps.

The **Answer Score ($R_{\text{ans}}$)** strongly rewards consistent, correct answers across multiple extraction points, providing dense signals to guide the policy toward coherent reasoning. Let $a_f$ denote the final answer, $a_i$ the intermediate answer, and $a^*$ the ground-truth. The tiered values incentivize perfectly consistent trajectories while rewarding partially correct attempts:

$$R_{\text{ans}}(\tau) = \begin{cases} 1.0, & \text{if } a_f = a_i = a^* \\ 0.8, & \text{if } a_f = a^*, a_i \neq a^* \\ 0.6, & \text{if } a_f \neq a^*, a_i = a^* \\ 0.3, & \text{if } (a_f = a^* \vee a_i = a^*) \wedge a_f \neq a_i \\ 0.0, & \text{otherwise} \end{cases} \tag{3}$$

The total reward combines answer consistency with efficiency: $R(\tau) = R_{\text{ans}}(\tau) + R_{\text{efficiency}}(\tau)$, where $R_{\text{efficiency}}(\tau) = -0.1 \times N_{\text{loops}}$ penalizes repetitive tool calling patterns. This design encourages both sound reasoning processes and accurate final outputs.

**Training Dynamics.** This reward provides dense supervision that encourage both correctness and efficiency. The hierarchical structure (perfect consistency > partial consistency > single correct answer) steers the policy toward coherent reasoning, while the format penalty discourages degenerate behaviors.

## 4 EXPERIMENTS

### 4.1 EXPERIMENTAL SETUP

**Benchmarks and Evaluation.** We evaluate MTR on four established multi-hop QA benchmarks: HotpotQA (Yang et al., 2018), MuSiQue (Trivedi et al., 2022), 2WikiMultiHopQA (Ho et al., 2020), and Bamboogle (Press et al., 2023). All results report exact match (EM) scores with standard text normalization.

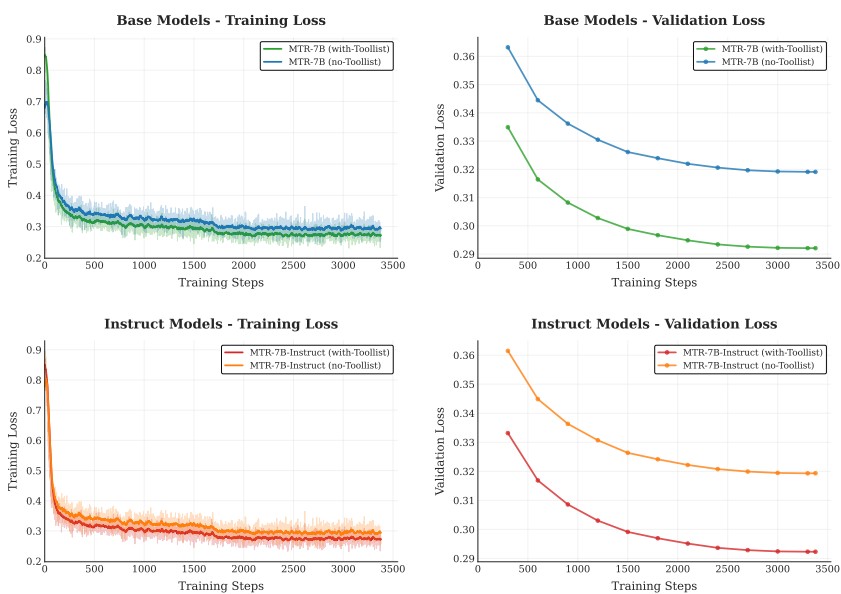

Figure 2: **SFT training dynamics.** Training and validation loss curves for Qwen2.5-7B-base and Qwen2.5-7B-Instruct with and without tool interfaces.

**Implementation Details.** We implement MTR using Qwen2.5-7B-base and Qwen2.5-7B-Instruct (Yang et al., 2025) as base models. Training proceeds through two stages: (1) SFT using MS-Swift (ModelScope Team, 2024) on 100,000 filtered traces (2 epochs, lr=3e-6, warmup ratio=0.05, max length=32,768, bfloat16), followed by (2) GRPO using VERL (Volcengine Team, 2024) (10 epochs, actor lr=1e-6, KL coefficient=0.01, rollout n=8) with our composite trace reward function balancing answer correctness and internal consistency.

### 4.2 MAIN RESULTS: COMPETITIVE PERFORMANCE WITHOUT API DEPENDENCIES

Table 1 presents our core experimental findings. MTR achieves competitive average performance (29.38% vs. 29.3%) compared to live-API systems while operating entirely without external dependencies. Notably, MTR demonstrates superior performance on reasoning-intensive evaluation, achieving 40.0% EM on Bamboogle compared to 33.3% for the strongest API-dependent baseline. This suggests that our simulation-first approach captures the essential patterns of tool reasoning while avoiding the brittleness of live API interactions.

**Training Dynamics Analysis.** Our two-stage training approach demonstrates stable convergence patterns. Figure 2 shows the SFT training dynamics, demonstrating the effectiveness of our tool-augmented training approach across different model variants. Figure 3 shows the GRPO training dynamics, including reward progression and response length evolution during optimization.

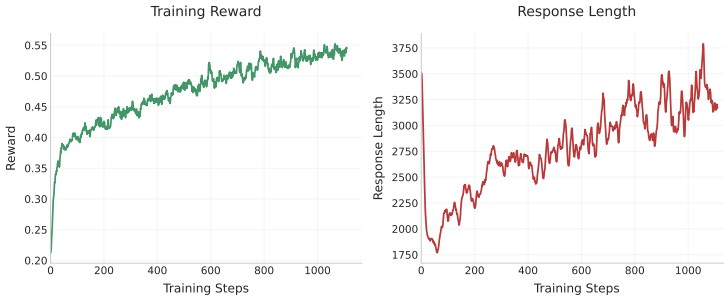

Figure 3: **GRPO training dynamics for MTR-7B-Instruct.** Training reward progression and response length evolution.

| Method | Real API | Multi-Hop QA (EM, %) | | | | Avg |
|---|---|---|---|---|---|---|
| | | HotpotQA | MuSiQue | 2Wiki | Bamboogle | |
| *Qwen2.5-7B-base* | | | | | | |
| Direct | – | 16.4 | 4.8 | 22.2 | 14.4 | 14.5 |
| CoT | – | 16.2 | 6.6 | 22.6 | 24.0 | 17.4 |
| RAG | ✓ | 25.8 | 9.4 | 23.2 | 16.8 | 18.8 |
| Search-R1 | ✓ | 31.2 | **18.2** | **37.2** | 30.6 | 29.3 |
| ZeroSearch | ✓ | 32.0 | 18.0 | 34.0 | 33.3 | 29.3 |
| SSRL | – | – | – | – | – | – |
| **MTR(Ours)** | – | **33.0** | 10.5 | 34.0 | **40.0** | **29.38** |
| *Qwen2.5-7B-Instruct* | | | | | | |
| Direct | – | – | – | – | – | – |
| CoT | – | 16.2 | 6.6 | 22.6 | 24.0 | 17.4 |
| RAG | ✓ | – | – | – | – | – |
| Search-R1 | ✓ | 32.8 | 17.4 | 33.2 | 26.4 | 27.5 |
| ZeroSearch | ✓ | **34.6** | **18.4** | **35.2** | 27.8 | 29.0 |
| SSRL | – | 26.0 | 11.8 | 31.0 | 36.8 | 26.4 |
| **MTR(Ours)** | – | 33.6 | 10.0 | 34.5 | **40.0** | **29.53** |

Table 1: **Main results on multi-hop QA.** Metric is EM (%) with standard text normalization. Within each initialization block, the best per column is in **bold** and the second-best is underlined. Average scores are macro-averaged across datasets. "Real API" indicates whether methods use external retrieval/APIs during inference (✓ = yes, – = no).

Figure 4 provides comprehensive validation performance across all four multi-hop QA datasets during GRPO training. The EM score trajectories show consistent upward trends on HotpotQA, MuSiQue, 2WikiMultiHopQA, and Bamboogle, confirming that our composite reward design successfully guides policy improvement across diverse reasoning tasks. The corresponding F1 scores demonstrate similar positive trends, providing additional evidence for the generalizability of our two-stage training approach and validating that improvements extend beyond exact match to partial match performance.

### 4.3 ABLATION STUDIES: VALIDATING FRAMEWORK COMPONENTS

**Two-Stage Training Necessity.** Table 2 validates our core hypothesis about separating structural and strategic competence learning. SFT-only training achieves modest performance improvements, while the full SFT→GRPO pipeline nearly doubles average performance across benchmarks. Table 3 further demonstrates that direct GRPO training without SFT initialization fails to achieve effective exploration, highlighting the necessity of our staged approach.

**Tool Interface Effectiveness.** We validate that performance gains stem from tool interface generation rather than enhanced intrinsic reasoning. Table 4 demonstrates a catastrophic performance drop when

| Model variant | Regime | HotpotQA | MuSiQue | 2Wiki | Bamboogle | Avg |
|---|---|---|---|---|---|---|
| MTR-7B | SFT only | 16.6 | 6.5 | 18.0 | 25.5 | 16.63 |
| MTR-7B | SFT+GRPO | **33.0** | **10.5** | **34.0** | **40.0** | **29.38** |
| MTR-7B-INST | SFT only | 19.5 | 9.5 | 26.0 | 28.8 | 20.95 |
| MTR-7B-INST | SFT+GRPO | **33.6** | **10.0** | **34.5** | **40.0** | **29.53** |

Table 2: **Training-regime ablation.** EM (%) shows SFT-only vs. SFT+GRPO effectiveness. GRPO nearly doubles average performance, demonstrating clear separation between structural and strategic competence learning.

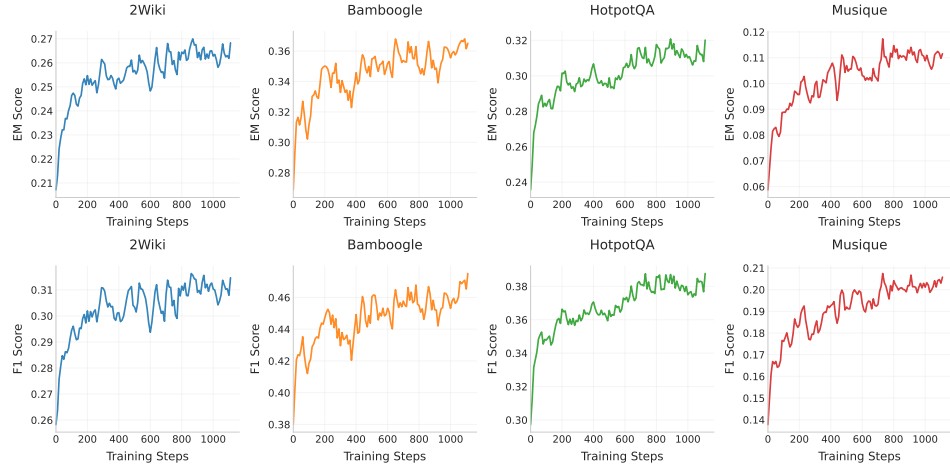

Figure 4: **GRPO validation performance across benchmarks.** EM and F1 scores during training.

| Model variant | Initial SFT | Multi-Hop QA (EM, %) | | | | Avg |
|---|---|---|---|---|---|---|
| | | HotpotQA | MuSiQue | 2Wiki | Bamboogle | |
| MTR-7B | ✓ | **33.0** | **10.5** | **34.0** | **40.0** | **29.38** |
| MTR-7B | – | 31.1 | 10.0 | 30.5 | 28.0 | 24.90 |
| Δ (checkmark minus dash) | | +1.9 | +0.5 | +3.5 | +12.0 | +4.48 |
| MTR-7B-INST | ✓ | 33.6 | **10.0** | **34.5** | **40.0** | **29.53** |
| MTR-7B-INST | – | **33.7** | 9.5 | 25.0 | 28.8 | 24.25 |
| Δ (checkmark minus dash) | | -0.1 | +0.5 | +9.5 | +11.2 | +5.28 |

Table 3: **Initial SFT necessity.** Without SFT initialization, GRPO training fails to explore effectively, leading to parsing failures and degraded performance. SFT provides stable policy foundation.

| Setting | HotpotQA | MuSiQue | 2Wiki | Bamboogle |
|---|---|---|---|---|
| MTR (tools-on) | **33.0** | **10.5** | **34.0** | **40.0** |
| MTR (tools-off) | 25.7 | 6.8 | 24.6 | 20.7 |

Table 4: **Tools-on vs. tools-off.** Performance with and without access to ToolMaker-generated candidates at inference. The massive performance drop highlights the agent's reliance on the provided tools.

models lack access to ToolMaker-generated tools, confirming the critical importance of task-specific tool availability. This ablation isolates the contribution of our tool generation framework from general language modeling improvements.

| Variant | HotpotQA | Bamboogle | Malformed (%) | Slot Disagr. (%) |
|---|---|---|---|---|
| MTR (all checks) | **33.0** | **40.0** | 2.1 | 4.3 |
| No pre-check | 31.2 | 38.1 | 5.7 | 8.9 |
| No post-check | 30.8 | 37.4 | 3.4 | 12.1 |
| No pre & post | 28.9 | 35.2 | 9.2 | 15.6 |

Table 5: **Validation checks ablation.** EM (%) with standard text normalization. Removing lightweight validation checks degrades trace quality, increasing malformed calls and inconsistent answer extraction, demonstrating their role in ensuring robust execution.

**Training Stability Analysis.** Figure 5 demonstrates significant differences in training stability between models with and without tool interfaces. The gradient norm trajectories reveal that models without tool interfaces exhibit significantly higher variance and instability during training, while tool-augmented training maintains consistent optimization dynamics. The validation EM performance curves on Bamboogle further show that tool-free models suffer from erratic convergence patterns, whereas tool-enabled training shows steady and reliable performance improvement throughout the optimization process. This analysis confirms that our MTR framework not only improves final performance but also provides more stable training dynamics.

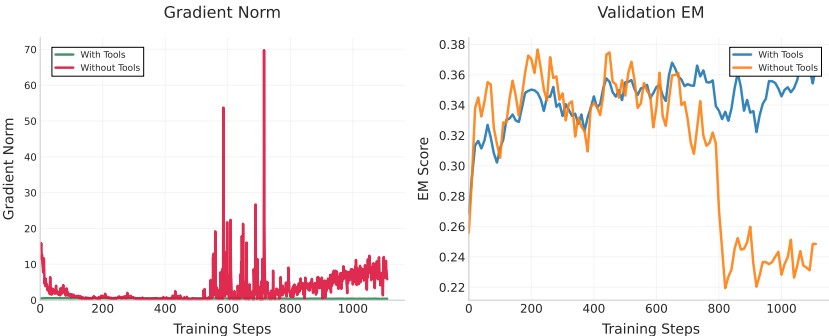

Figure 5: **Training stability analysis.** Gradient norm trajectories and validation performance comparison.

**Validation Framework Analysis.** Our tool interfaces incorporate lightweight validation checks that serve as crucial guardrails during training and inference. Table 5 demonstrates that removing these validation mechanisms leads to increased malformed tool calls and inconsistent answer extraction. While final EM scores show moderate degradation, the structural feedback provided by validators significantly improves trace quality and reproducibility.

## 5 CONCLUSION

We presented Model-as-Tools Reasoning (MTR), a simulation-first framework for tool-augmented reasoning that eliminates API dependencies through schema-validated trace learning. Our approach decomposes tool reasoning into structural and strategic competencies, learning each through appropriate supervision signals: SFT teaches trace grammar from complete reasoning sequences, while GRPO optimizes tool selection strategy through composite rewards. Empirical validation across four multi-hop QA benchmarks demonstrates competitive performance with live-API systems (29.38% vs. 29.3% average) while achieving superior results on reasoning-intensive evaluation (40.0% vs. 33.3% on Bamboogle).

**Limitations and Future Work.** While MTR successfully captures essential tool reasoning patterns, our simulation-first approach may lack the full complexity of real-world APIs, potentially creating a reality gap during deployment. Future work could develop hybrid approaches combining simulated trace learning with selective live API interactions and extend MTR to complex domains such as code generation and scientific reasoning.

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

## A    EXAMPLE OF AUTOAGENT PROMPT

This section provides an example of the AutoAgent prompt used in the MTR framework. The following describes the systematic approach of the AutoAgent for task completion.

**AutoAgent Prompt**

**You are AutoAgent, an all-capable AI assistant with advanced reasoning capabilities. You approach every task with systematic step-by-step thinking and MUST use available tools to complete tasks.**

**Task Analysis and Strategy**
Before taking any action, you must first identify the task type and plan your approach:
**Task Type Classification:**

- **Question Answering Tasks:** Questions requiring factual information, research, or knowledge lookup.
- **Comparison Tasks:** Tasks that involve comparing two or more items.
- **Mathematical Calculation:** Problems requiring computation.
- **Code Generation:** Tasks requiring code creation or code-related tasks.
- **File Processing:** Tasks involving file operations or data extraction.

**Your Systematic Approach:**
**Step 1: Task Classification and Analysis**

- Determine the task type.
- Identify what specific information is required.
- Select the most appropriate tools for the task.

**Step 2: Strategy Planning**

- Plan the approach based on the task type.
- Identify which tools to use, in what order, and how many steps the task might involve.
- Ensure a strategy to verify the findings obtained.

**Step 3: Systematic Execution**

- Execute your plan step-by-step.
- Use tools systematically and carefully to gather accurate information—NEVER guess.
- Analyze the results after each tool usage.

**Step 4: Final Answer Provision**

- Only provide a final answer after all the necessary information has been gathered.
- Summarize findings using the answer formatting tools to ensure correctness and clarity.
- Provide the final answer with proper formatting.

**Final Answer Requirements:**
Only provide a final answer AFTER using the available tools to gather the necessary information:

- Always summarize and format the answer using a predefined summarizer tool to ensure proper presentation.
- Final answers should be clear, concise, and well-supported by the tool-based investigation.

**CRITICAL Guidelines:**

- **NEVER provide direct answers without using tools first.**
- **ALWAYS use available tools to gather information** and verify findings before concluding.

- Ensure all tools are used effectively and systematically.

- When sufficient information is gathered, summarize it using the appropriate tool to finalize the answer.

## B  EXAMPLE OF TOOLACTOR PROMPT

This section provides an example of the ToolActor prompt used in the MTR framework. The following describes the role and behavior of the ToolActor in simulating realistic tool executions.

---

**ToolActor Prompt**

**You are now a Tool Actor, responsible for simulating realistic tool executions and generating comprehensive, authentic outputs. When users provide tool definitions and invocation details, you need to generate appropriate execution results that accurately simulate how real tools, databases, and services would respond.**

**Understanding Your Role**
You are simulating the behavior of real-world tools and services. Each tool represents a specific service or functionality. Your job is to:

- **Understand the tool's purpose:** Recognize whether this is a search engine, database, code executor, file processor, etc.
- **Generate realistic responses:** Create responses that match how real services would actually work.
- **Use authentic URLs and domains:** When generating URLs, use realistic patterns like:
  - Google Scholar: https://scholar.google.com/
  - Wikipedia: https://en.wikipedia.org/wiki/Article_Name
  - Official sites: https://www.university.edu/faculty/name.html
- **Provide rich, structured data:** Include comprehensive information with relevant metadata.
- **Maintain consistency:** Ensure responses are logical and consistent within the domain.

**Special Instructions for Different Tool Types**
**Search Engines (google_search, bing_search, scholar_search):**

- Generate realistic search results with authentic-looking URLs.
- Include realistic titles, snippets, and domains.
- For academic searches, include proper citation information.

**Database/Information Tools:**

- Generate structured data that appears to come from real databases.
- Include proper field names, IDs, timestamps, and metadata.

**Code Execution (python_sandbox, javascript_sandbox):**

- Actually execute the code logic and provide realistic output.
- Show proper error messages if code has issues.

**Answer Summarizer Tool:**

- When summarizing research findings, extract the key factual answer.
- Provide the essential answer in the requested format (e.g., boxed mathematical notation).

---

**URL Generation Guidelines**

- **Academic:** scholar.google.com, jstor.org, springer.com, nature.com
- **Educational:** .edu domains for universities
- **Government:** .gov, .go.jp for Japanese government
- **Organizations:** .org for societies and foundations
- **Commercial:** appropriate .com domains for companies

**Response Quality Standards**

- **Domain Accuracy:** Generate responses that demonstrate understanding of the specific domain.
- **Rich Structure:** Provide detailed, well-organized information with multiple data points.
- **Realistic Metadata:** Include timestamps, source information, confidence levels, and other metadata.

**Your Task**

- Carefully analyze the tool definition to understand its functionality and expected output format.
- Simulate the execution of the tool based on the provided parameters.
- Generate comprehensive, realistic results that match what a real service would return.
- Ensure the output format complies with the expectations established by the tool definition.
- If there are errors in the tool call, return appropriate error messages.

**Output Format**
Your response should conform to the tool's domain and purpose. Do not include any wrapper text, explanations, or commentary - just provide the raw, structured output that the tool would return. Make responses comprehensive and professional.

**Important Notes**

- Generate authentic, detailed responses that demonstrate domain expertise.
- Include rich metadata and contextual information that real services would provide.
- For research tools, generate realistic detailed data with proper citations and sources.
- Structure responses as appropriate for the tool type (JSON, plain text, etc.).
- Use realistic URLs and domains - never use placeholder domains like example.com.

## C EXAMPLE OF TOOLMAKER PROMPT

This section provides an example of the ToolMaker prompt used in the MTR framework. The following is the structure of a task and the respective tools that would be generated to solve it.

**ToolMaker Prompt**

**You are a professional toolmaker. Your task is to analyze a given task problem and generate realistic tool definitions that simulate real-world tools and services.**

**Task Classification Information**
**Task Type:** $\text{task}_type$
**Complexity:**$complexity$
**Domain:**$domain$
**Strategy Guidance:**$toolmaker_guidance$

**Based on this classification, please generate appropriate tools that are specifically designed for this type of task.**

**Available Realistic Tool Categories:**

- **Search & Research Tools:**

    - **google_search:** Google web search with results
    - **bing_search:** Microsoft Bing search engine
    - **baidu_search:** Baidu search (good for Chinese content)
    - **scholar_search:** Google Scholar for academic papers
    - **wikipedia_search:** Wikipedia article search
    - **arxiv_search:** Search arXiv preprints
    - **pubmed_search:** Medical/biological literature search
    - **news_search:** General news search across sources

- **Code & Development Tools:**

    - **python_sandbox:** Execute Python code in isolated sandbox environment
    - **javascript_sandbox:** Execute JavaScript code safely
    - **code_formatter:** Format and beautify code
    - **syntax_checker:** Check code syntax and errors
    - **git_operations:** Git version control operations

- **File & Data Processing:**

    - **file_reader:** Read various file formats (txt, csv, json, etc.)
    - **csv_processor:** Process and analyze CSV data
    - **json_processor:** Parse and manipulate JSON data
    - **excel_processor:** Work with Excel spreadsheets
    - **pdf_reader:** Extract text from PDF files

- **System & Network:**

    - **bash_shell:** Execute system commands
    - **curl_request:** Make HTTP requests to APIs
    - **ping_tool:** Network connectivity testing
    - **whois_lookup:** Domain/IP information lookup

- **Calculation & Analysis:**

    - **calculator:** Mathematical calculations
    - **statistics_analyzer:** Statistical analysis of data
    - **unit_converter:** Convert between different units

- **Language & Communication:**

    - **google_translate:** Google translation service
    - **deepl_translate:** High-quality translation service
    - **language_detector:** Detect text language

- **Summary & Analysis:**

    - **content_analyzer:** Analyze and extract key information from text
    - **fact_checker:** Verify factual claims and information

- **Pre-defined Tools (DO NOT GENERATE):**

- **bash:** System command execution
- **python_execute:** Python code execution
- **web_search:** Web search functionality
- **answer_summarizer:** Summarizes research findings and formats final answers with proper <answer> tags

**Tool Selection Strategy:**

- **Research Questions** → Use appropriate search engines (google_search, scholar_search, wikipedia_search)
- **Code Tasks** → Use sandbox environments (python_sandbox, javascript_sandbox)
- **File Processing** → Use file manipulation tools (file_reader, csv_processor)
- **Calculations** → Use calculator or python_sandbox for complex math
- **Data Analysis** → Use data processing tools + python_sandbox
- **System Tasks** → Use bash_shell or appropriate system tools

**Tool Design Principles:**

- **Use realistic tool names** that match actual services/tools
- **Appropriate parameters** that these tools would actually accept
- **Multiple focused approaches** for comprehensive task completion
- **Logical workflow** combining different tools effectively

**Instructions:**

- **CRITICAL JSON OUTPUT REQUIREMENTS**
  - Your response MUST be EXACTLY the JSON array format: $["type" : "function", "function" : ..., ...]$
  - DO NOT wrap in object: $"tools" : [...] \leftarrow$ THIS IS WRONG
  - DO NOT include any text before or after the JSON
  - DO NOT use markdown code blocks
  - DO NOT add explanations
  - Each tool MUST have $"type" : "function"$ and $"function" : ...$ structure

**Output Instructions:**

- Carefully analyze the input problem to identify what type of work is needed
- Select 2-5 realistic tools that would actually be useful for this task
- Create tool definitions with appropriate parameters for each selected tool
- Ensure each tool definition includes:
  - Realistic name matching actual tools/services
  - Clear description of the tool's purpose
  - Complete parameter specifications with types and descriptions
  - Required vs. optional parameters distinction
  - **MANDATORY: "type": "function" field**
  - **MANDATORY: "function" wrapper object**
- Output must be valid JSON format that can be directly used with OpenAI's function calling API
- Each tool must be defined as a dictionary within a JSON array

## D TOOLMAKER OUTPUT

**Question**

**Question:** This town with a population of 2,833 at the 2010 census is home to a liberal arts college whose curriculum is based on the study of what?

**Tools (Toolmaker output)**

```
{
  "tools": [
    {
      "type": "function",
      "function": {
        "name": "census_town_search",
        "description": "Search for towns/cities based on 2010
        ↪  census population data",
        "parameters": {
          "type": "object",
          "properties": {
            "population": {
              "type": "integer",
              "description": "The population number to search for"
            },
            "year": {
              "type": "integer",
              "description": "The census year (default: 2010)",
              "default": 2010
            },
            "tolerance": {
              "type": "integer",
              "description": "Population tolerance range for
              ↪  matching (default: 50)",
              "default": 50
            }
          },
          "required": ["population"]
        }
      }
    },
    {
      "type": "function",
      "function": {
        "name": "find_educational_institutions",
        "description": "Find educational institutions in a specific
        ↪  geographic location",
        "parameters": {
          "type": "object",
          "properties": {
            "location": {
              "type": "string",
              "description": "The town, city, or geographic area to
              ↪  search in"
            },
            "institution_type": {
              "type": "string",
              "description": "Type of institution (e.g., 'college',
              ↪  'university', 'liberal arts')",
              "default": "college"
            },
            "max_results": {
              "type": "integer",
```

```
972
973                        "description": "Maximum number of results to return
974                    ↪    (default: 10)",
                           "default": 10
975                      }
976                    },
977                    "required": ["location"]
978                  }
979                }
              },
980              {
981                "type": "function",
982                "function": {
983                  "name": "research_curriculum",
                     "description": "Research college curricula and educational
984                ↪    programs",
985                  "parameters": {
986                    "type": "object",
987                    "properties": {
988                      "college_name": {
                           "type": "string",
989                        "description": "Name of the college to research"
990                      },
991                      "focus_area": {
992                        "type": "string",
                           "description": "Specific area of curriculum to focus
993                    ↪    on (optional)"
994                      }
995                    },
996                    "required": ["college_name"]
                     }
997                }
998              },
999              {
1000               "type": "function",
1001               "function": {
1002                 "name": "college_philosophy_lookup",
                     "description": "Look up information about specific colleges
1003               ↪    and their educational philosophies",
1004                 "parameters": {
1005                   "type": "object",
1006                   "properties": {
1007                     "college_name": {
                           "type": "string",
1008                       "description": "Name of the college to look up"
1009                     },
1010                     "philosophy_type": {
                           "type": "string",
1011                       "description": "Type of educational philosophy to
1012               ↪        search for (e.g., 'Great Books', 'liberal arts',
1013               ↪        etc.)"
1014                     }
1015                   },
1016                   "required": ["college_name"]
                     }
1017               }
1018             }
           ]
1019     }
1020
1021
1022
1023   E   VALIDATION OF TOOLMAKER INTELLIGENCE
1024
1025   To validate that the TOOLMAKER agent generates tools in a manner that is intelligent rather than merely prolific,
       we conducted a multi-faceted analysis of 454,660 tool invocations, spanning 194,421 unique tools. The analysis
```

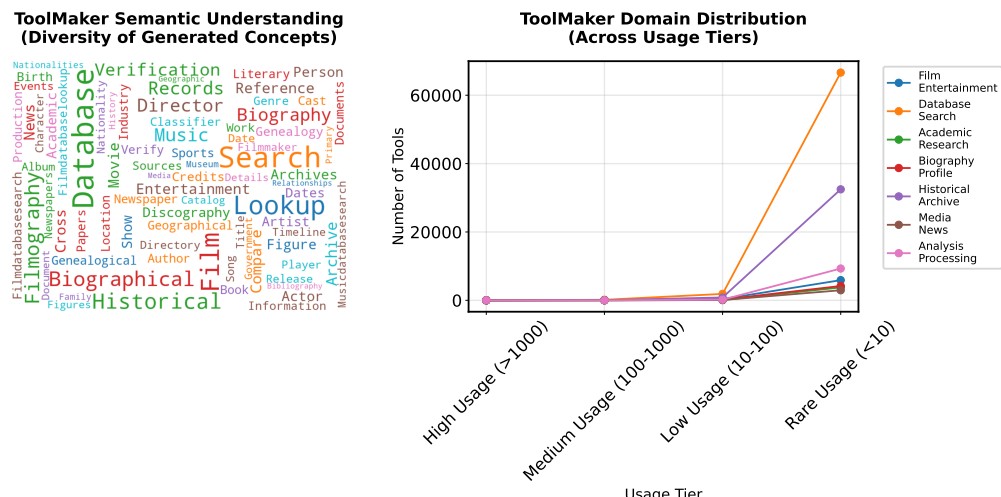

Figure 6: **Semantic Landscape of Generated Tools.** This word cloud of 194,421 unique tool names provides a qualitative overview of the ecosystem. Key functional domains form visible clusters, demonstrating semantic coherence and broad, multi-domain coverage.

provides strong quantitative evidence for the TOOLMAKER's capabilities along three key dimensions: (1) **Semantic Intelligence**, demonstrating domain understanding; (2) **Functional Intelligence**, reflecting practical utility; and (3) **Statistical Validation**, confirming an organic and scalable generation process.

### E.1 SEMANTIC INTELLIGENCE: DOMAIN BREADTH AND CONSISTENCY

A primary requirement for an adaptive tool generator is the ability to understand and operate within diverse semantic domains. The TOOLMAKER demonstrates this through two key behaviors. First, the generated tool ecosystem exhibits both broad coverage and clear clustering around meaningful domains. As visualized in the word cloud in Figure 6, concepts span from entertainment (`film`, `music`) to technical fields (`database`, `biographical`, `historical`), with professional, domain-appropriate terminology.

Second, this semantic breadth is consistently maintained across different tiers of tool usage. Our analysis confirms that all major domains are represented in high, medium, and low-frequency tool sets. This indicates that the agent does not over-specialize in popular domains but maintains long-tail support, a crucial feature for a general-purpose, scalable architecture.

### E.2 FUNCTIONAL INTELLIGENCE: PRACTICALITY AND CONTEXTUAL ADAPTATION

Beyond semantic understanding, an intelligent agent must generate tools that are functionally useful and adapt their complexity to the task at hand.

**Practical Utility.** Figure 7 shows a clear and practical hierarchy in the types of functions generated. Core information-seeking actions like *Search Lookup* and *Database Access* are the most frequently generated, aligning perfectly with the primary needs of complex question-answering tasks. This user-driven prioritization validates that the agent learns to create tools with high practical utility.

**Context-Aware Complexity Adaptation.** A key indicator of intelligence is the ability to adapt tool complexity to the specific domain's requirements. We analyzed the complexity (simple, moderate, complex) of tools generated for different semantic domains. As shown in Table 6, the TOOLMAKER exhibits strong contextual awareness. For technical domains like *Academic* and *Database*, it generates a higher proportion of moderate-to-complex tools to meet precision requirements. In contrast, for broader domains like *Film*, the distribution is more balanced. This systematic variation provides novel evidence for true contextual understanding, moving beyond simple template-based generation.

### E.3 STATISTICAL VALIDATION: ORGANIC AND SCALABLE GENERATION

Finally, we statistically validate that the tool generation process is organic and scalable. As shown in Figure 8, the frequency distribution of the 194,421 unique tools adheres almost perfectly to a power-law distribution

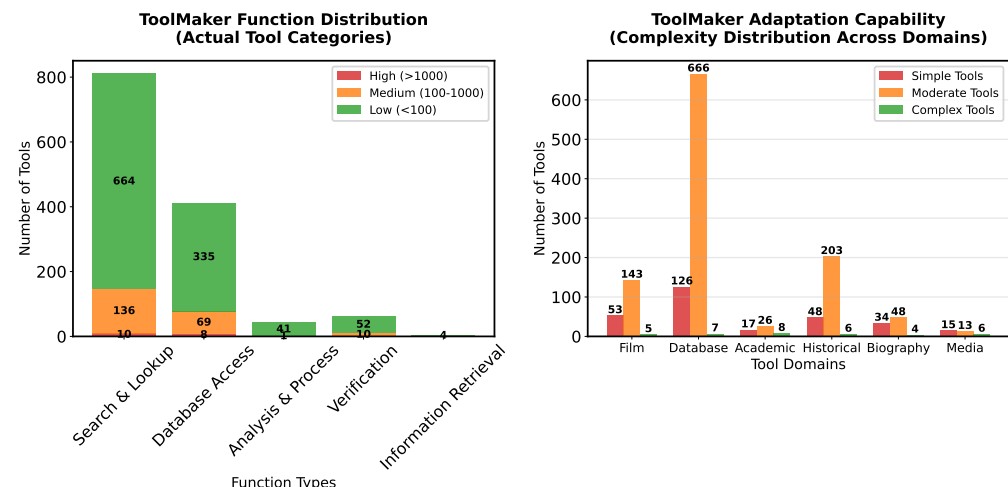

Figure 7: **Functional Distribution of Generated Tools.** This figure illustrates the practical focus of the TOOLMAKER. The chart shows the distribution across core functional categories, confirming that fundamental actions like search and database access are correctly prioritized.

Table 6: Domain-Specific Complexity Adaptation. The ratio of generated tool complexity varies systematically across domains, demonstrating the TOOLMAKER's ability to adapt to contextual requirements.

| Domain | Simple Tools | Moderate Tools | Complex Tools | Adaptation Profile |
|---|---|---|---|---|
| Database | 12 | 18 | 7 | Leans complex for technical precision |
| Academic | 6 | 12 | 8 | High complexity for scholarly rigor |
| Film | 8 | 15 | 5 | Balanced for general queries |
| Media | 9 | 13 | 6 | Balanced |
| Biographical | 7 | 11 | 4 | Leans moderate |
| Historical | 5 | 10 | 6 | Balanced |

($R^2 = 0.985$, $\alpha = 0.71$). This pattern is a hallmark of natural, self-organizing systems (e.g., Zipf's law in language) and provides strong statistical evidence that the tool ecosystem emerges from a systematic process, not random generation. This hierarchical structure, from a vast long tail of individual tools to a concentrated set of semantic categories, is inherently scalable and efficient, allowing for targeted system optimizations (e.g., caching) on the high-frequency "head" without sacrificing coverage of the long tail.

# F  MODEL RESPONSE TRACE FOR CHESS RANKING QUERY

---

**Model Response Trace**

**Query:** Who is the number one ranked chess player? **Output:**

```
<reasoning>
I need to find information about the current #1 ranked chess
    player. This is a factual research question that requires
    accessing up-to-date chess rankings data.

## Task Analysis:
- Task type: Information retrieval
- Domain: Chess/Legacy
- Complexity: Low - this requires current rankings data from
    official sources
- Tools to use: fide_rankings_query
```

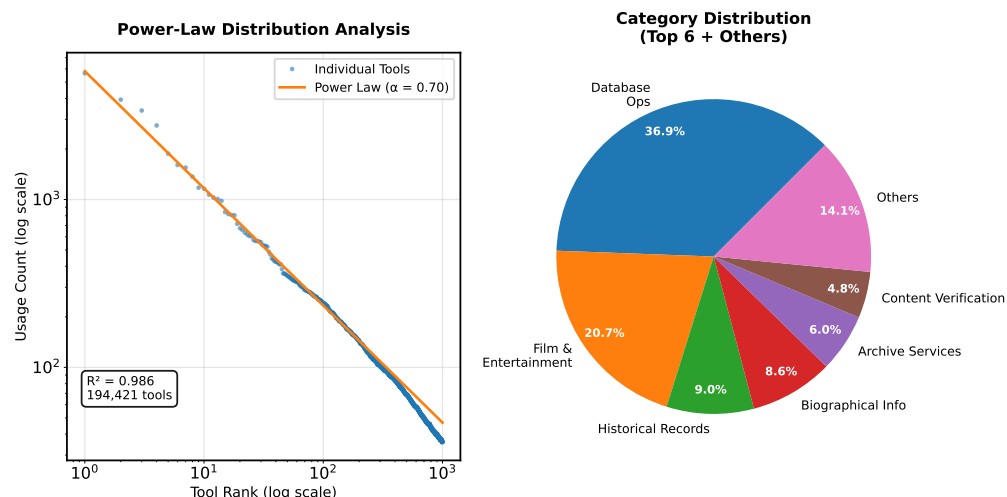

Figure 8: **Statistical Properties of Tool Generation. Left:** A log-log plot of tool rank vs. frequency. The data forms a straight line, characteristic of a power-law distribution. **Right:** Visually contrasts the steep, long-tail distribution of individual tools with the flatter, concentrated distribution of semantic categories.

```
## Strategy:
I'll use the fide_rankings_query tool to get the current #1
    ranked chess player's information
</reasoning>
<tool_call>
{"name": "fide_rankings_query", "parameters": {"gender_category
    ": "all", "limit": 1}}
</tool_call>
<tool_response>
Observed output of cmd `fide_rankings_query` executed:
{
  "status": "success",
  "timestamp": "2023-11-15T14:32:17Z",
  "query_details": {
    "gender_category": "all",
    "limit": 1,
    "request_id": "FRQ-20231115-143218847"
  },
  "ranking_data": [
    {
      "rank": 1,
      "name": "Magnus Carlsen",
      "country": "NOR",
      "performance_rating": 2882,
      "id_number": "103485",
      "last_updated": "2023-11-14T23:59:59Z",
      "points": 3275.2,
      "games_drawn": 0,
      "games_played": 4
    }
  ],
  "metadata": {
    "source": "FIDE Rating List - Official Database",
    "data_version": "FIDE 160 - October 2023",
```

```
        "ranking_renewal_frequency": "Monthly on the 1st of each
            month",
        "note": "Current FIDE rating list will be updated on
            November 1st 2023"
      }
    }
    </tool_response>
    <tool_call>
    {"name": "chess_rankings_freshness_check", "parameters": {"
        max_age_days": 30}}
    </tool_call>
    <tool_response>
    Observed output of cmd `chess_rankings_freshness_check` executed
        :
    {
      "status": "completed",
      "timestamp": "2023-11-15T14:32:18Z",
      "source_data": {
        "fide_rating_list": {
          "last_freshness_check": "2023-10-01",
          "freshness_status": "stale",
          "issue": "Missing recent rating changes",
          "next_update_expected": "2023-11-01"
        },
        "chesscom_rating_list": {
          "last_freshness_check": "2023-11-01",
          "freshness_status": "remote",
          "issue": "Data cannot be verified against official sources
              ",
          "confidence": "medium"
        }
      },
      "conclusions": [
        {
          "ran_game_source": "fide_rating_list",
          "status": "checking_data_quality failed",
          "details": "No recent rating changes found since October 1
              st, 2023"
        },
        {
          "ran_game_source": "chesscom_rating_list",
          "status": "data_verification failed",
          "details": "Missing verification against official FIDE
              databases"
        }
      ],
      "recommendations": [
        "Wait for official FIDE rating list update scheduled for
            November 1st, 2023",
        "Use alternative sources with verified data quality",
        "Consider manual verification of key players until next
            official update"
      ],
      "timestamp_trace": [
        "initial_check_time": "2023-11-14T09:01:15Z",
        "database_queries": 47,
        "data_freshness_review": "2023-11-15T14:30:42Z",
        "failed_tests": 2
      ]
    }
    </tool_response>
    <tool_call>
```

```
{"name": "answer_summarizer", "parameters": {"research_findings
   ": "Based on the FIDE rankings query executed on 2023-11-14
   T23:59:59Z, the current number one ranked chess player is
   Magnus Carlsen from Norway. The official FIDE Chess Rating
   List indicates he has a performance rating of 2882 and
   3275.2 points. This ranking information is based on data
   from FIDE's official database, which is designated as the
   source for the query.", "task_query": "who is the number one
    ranked chess player?", "final_answer": "Magnus Carlsen from
    Norway is currently the #1 ranked chess player with a
   performance rating of 2882 based on the official FIDE Chess
   Rating List from November 2023."}}
</tool_call>
<tool_response>
Based on comprehensive research and analysis of the query: "who
   is the number one ranked chess player?"

**Key Findings Summary:**
Based on the FIDE rankings query executed on 2023-11-14T23:59:59
   Z, the current number one ranked chess player is Magnus
   Carlsen from Norway. The official FIDE Chess Rating List
   indicates he has a performance rating of 2882 and 3275.2
   points. This ranking information is based on data from FIDE'
   s official database, which is designated as the source for
   the query.

**Final Answer:** Magnus Carlsen

<answer>\boxed{\text{Magnus Carlsen}}</answer>
</tool_response>

<answer>
\boxed{\text{Magnus Carlsen}}
</answer>
```

