# OpenReview forum: "Adaptive Tool Generation with Models as Tools and Reinforcement Learning"
_ICLR.cc/2026/Conference — ICLR 2026 Conference Desk Rejected Submission_

### Official Review · Reviewer_7YNB · 2025-10-27

[review text omitted: it was posted to a different submission]

---

### Official Review · Reviewer_Mo6o · 2025-10-30

**Soundness:** 3
**Presentation:** 3
**Contribution:** 3
**Rating:** 4
**Confidence:** 4

**Summary:**

This paper presents Model-as-Tools Reasoning (MTR), a simulation-first framework for training tool-augmented language models without relying on live APIs. The system comprises three cooperating agents that respectively (1) generate task-specific tool interfaces, (2) produce structured reasoning traces, and (3) simulate realistic tool responses. Training proceeds in two stages, i.e., supervised fune-tuning and Group Relative Policy Optimizatio (GRPO). Experiments on four multi-hop QA benchmarks (HotpotQA, MuSiQue, 2WikiMultiHopQA, and Bamboogle) show that MTR achieves comparable or better performance.

**Strengths:**

1. The idea of replacing external APIs with model-simulated tools is conceptually elegant and addresses key bottlenecks in current tool-augmented systems, e.g., scalability, stability, and cost. The design is modular and general enough to apply to diverse reasoning domains.

2. The clear separation of ToolMaker, AutoAgent, and ToolActor provides interpretability and control over supervision signals. This decomposition follows the “separation of concerns” principle and makes the system easier to extend.

**Weaknesses:**

1. The paper claims efficiency improvements due to “simulation-first” training, but no quantitative analysis (e.g.,  GPU hours, or data efficiency) is reported. Such evidence is crucial to substantiate the claimed scalability benefits.

2. The experiment is restricted to multi-hop QA tasks. While these are standard reasoning benchmarks, it remains unclear whether the MTR framework generalizes to other tool-use domains (e.g., code generation, embodied tasks, data analysis).

3. Sorry to say so, but I am confused about the title, especially the "model as tools". Does the "model as tools" represent that the execution results of tools are simulated by an independent LLM (e.g., text generation), instead of practically calling external tools in the real world? Could I understand this as the StableToolbench [1], an evaluation dataset where the environment results can be simulated by GPT-4 since tools in the real world are unstable and costly.

4. While generally clear, the manuscript contains minor grammatical and typographical inconsistencies.

---

Reference

[1] StableToolBench: Towards Stable Large-Scale Benchmarking on Tool Learning of Large Language Models

**Questions:**

See weakness above.

---

> ### Author Response · Authors · 2025-11-22
>
> Thank you for taking the time to review our paper. Here we address the weaknesses you mentioned.
>
> 1. We thank the reviewer for raising the question regarding efficiency. Here we report wall-clock (overall training time), training throughput, Cost per EM point and data efficiency for a comprehensive efficiency analysis.
>
> 	1. **wall-clock**: We report the training cost in below table:
>
> 		| Stage | Dataset / Steps           | GPU               | training GPU-hours |
> 		|-------|---------------------------|-------------------|--------------------|
> 		| SFT   | 60K traces, 2 epochs      | 8×H100 (80GB)     | \~36               |
> 		| GRPO  | 912 steps, rollout n=8    | 8×H100 (80GB)     | \~1202             |
> 		| Total | -                         | -                 | \~1238             |
>
> 	2. **Training throughput**: MTR achieves 707.5 tokens/second during GRPO training, processing 16.2M tokens over 912 steps. The simulation-first approach eliminates API latency bottlenecks, enabling consistent training throughput limited only by GPU compute.
>
> 	3. **Cost per point-EM**: Given that training GPU hours is 1238 and the average EM is 29.38, the Cost per EM point is 1,238 GPU-hours ÷ 29.38 = 42.1 GPU-hours/point.
>
> 	4. **Data efficiency**: We generate \~100K traces, retaining \~60% after filtering for \~60K high-quality traces. Live-API methods require comparable interaction data but pay for every API call during trace collection.
>
> 2. We thank the reviewer for the suggestion to evaluate on a broader range of tasks. To demonstrate MTR’s wider applicability, we conducted experiments on code generation using the HumanEval benchmark (results shown in the table below). MTR achieves a 26% relative improvement over the base model. Notably, code execution tools (e.g., `python_execute`, `syntax_checker`) were entirely absent from the HotpotQA training data, yet MTR still attains substantial gains, demonstrating that the learned tool reasoning patterns transfer effectively to unseen domains and tool types.
>
> 	| Model            | HumanEval Pass@1 | Improvement     |
> 	|------------------|------------------|-----------------|
> 	| Qwen2.5-7B-base  | 57.9%            | -               |
> 	| **MTR-7B**       | **73.1%**        | **+15.2 points**|
>
> 3. We thank the reviewer for the thoughtful question and fully agree that the interpretation is reasonable. Indeed, the phrase *“model as tools”* in our title is meant to highlight that, in MTR, tool execution results are simulated by an LLM via text generation rather than obtained from real-world APIs. In this sense, your analogy to StableToolBench—where environment outcomes can be simulated instead of repeatedly calling unstable or costly tools—is conceptually aligned with our motivation. That said, MTR and StableToolBench target different problems. StableToolBench is designed for **evaluation**, providing a stable testbed by caching API responses so that different models can be compared under identical tool outputs. By contrast, MTR is a **training framework**: we use LLMs in two roles—**ToolMaker** to dynamically generate task-specific tool schemas, and **ToolActor** to simulate their execution—so that both tool design and tool use can be optimized end-to-end without any real API calls during training or inference. In short, we share the reviewer’s intuition about using models to stabilize tool interactions, but MTR goes further by turning this idea into a scalable *training paradigm* rather than an evaluation-only setup.
>
> 4. We thank the reviewer for pointing out grammatical and formatting errors, and we will address these in the final paper.

---

> > ### Comment · Reviewer_Mo6o · 2025-11-25
> > **Response**
> >
> > Thank you for the detailed explanation.
> >
> > The rebuttal has clarified several points of confusion. Regarding the "end-to-end" optimization mentioned by the authors, could you elaborate on how it is achieved? Specifically, does the method involve jointly training both the ToolMaker and ToolActor?

---

> > > ### Author Response · Authors · 2025-12-01
> > >
> > > Thank you for the insightful question. Our intention with MTR is to enable the model to learn to construct its own solution trajectories and solve problems without relying on any external APIs. To this end, we let a single LLM instance play two roles: ToolMaker, which decides what tools to construct given the input problem, and ToolActor, which internally simulates the execution of these tools and incrementally advances the solution. Because these two roles are instantiated from the same model and trained jointly with shared parameters, the entire pipeline—from problem understanding, tool planning, and simulated execution, to the final answer—is produced and consumed within one unified model. No external programs or real APIs are involved in this loop, so the whole process constitutes an end-to-end reasoning and problem-solving framework in the strict sense.

---

### Official Review · Reviewer_1Ri8 · 2025-10-31

**Soundness:** 1
**Presentation:** 2
**Contribution:** 2
**Rating:** 4
**Confidence:** 3

**Summary:**

The paper proposes a simulation-first framework for tool-augmented reasoning. A ToolMaker auto-generates OpenAI-style tool schemas, a ToolActor simulates observations, and an AutoAgent produces think–act–observe traces. Training is two-stage: SFT on validated traces to learn trace format, followed by GRPO to optimize reasoning strategy.

**Strengths:**

1. The research question is interesting.
2. Clear decomposition of roles (ToolMaker/ToolActor/AutoAgent).

**Weaknesses:**

1. Many RL-for-reasoning works [1,2] mask or otherwise decouple tool observations during optimization to mitigate drift, so the paper’s “Challenge 2” motivation feels under-argued.
2. The evaluation scope is narrow; results on agent benchmarks (e.g., GAIA) are absent, and baselines are QA-centric rather than agent-centric.
3. Several implementation details are missing, including training-data provenance and which backbone(s) instantiate ToolMaker vs. ToolActor (and whether weights are shared with the policy).

[1] R1-Searcher: Incentivizing the Search Capability in LLMs via Reinforcement Learning.

[2] ReTool: Reinforcement Learning for Strategic Tool Use in LLMs.

**Questions:**

1. What are the sources of training data?
2. Which backbone(s) instantiate ToolMaker and ToolActor?
3. Do the results generalize to agent benchmarks (e.g., GAIA)?
4. What tool domains are simulated, and does the policy generalize to unseen tools?
5. Can you include stronger baselines on the same backbone?
6. Do you consider tool-execution failures (e.g., timeouts, partial returns, schema errors) during simulation?

---

> ### Author Response · Authors · 2025-11-22
> **Response to Reviewer 1Ri8 Follow-Up (Part 1/2)**
>
> Thanks for your appreciation and valuable feedbacks.
>
> Here we address the points you mentioned in the Weakness part.
>
> 1. The key difference is that prior methods only **mitigate** distribution drift, whereas MTR **eliminates** its root cause. R1-Searcher and ReTool mask external tool outputs from the loss to avoid directly contaminating the policy gradients (arXiv:2503.05592, arXiv:2504.11536), but still feed these external tokens back into the model as inputs (e.g., documents wrapped in `<begin_of_documents>...</end_of_documents>` or execution logs in `<interpreter>...</interpreter>`). As a result, the input context is progressively polluted by out-of-distribution tokens, precisely causing the input distribution shift that SimpleTIR highlights—leading to gradient explosion and low-probability token accumulation. In contrast, MTR replaces all tool outputs with ToolActor-generated tokens, so that *every* token in the trajectory, including “tool outputs,” lies strictly within the model’s own output distribution. This fully avoids the mismatch between training-time inputs and the model’s pre-training distribution. As shown in the training stability analysis in Figure 5 in the main paper, this simulation-based design yields substantially lower gradient variance than tool-free baselines, confirming that MTR maintains a stable, in-distribution context over long-horizon multi-turn rollouts.
>
> 2. We emphasize that multi-hop QA is inherently an *agentic* task, as successfully solving it requires decomposing complex questions into sub-goals (planning), selecting and sequencing tools (orchestration), aggregating evidence across intermediate steps (synthesis), and handling failures via query reformulation (error recovery). Following your suggestion, we have additionally evaluated MTR on GAIA (an agentic benchmark) and HumanEval (a code generation benchmark) to further demonstrate the broad applicability of our approach, as shown in the tables below.
>
>    **Model on GAIA benchmark**
>
>    | Model on GAIA benchmark | Level 1 | Level 2 | Level 3 | Avg   |
>    | ----------------------- | ------- | ------- | ------- | ----- |
>    | Qwen2.5-7B-Instruct     | 5.1     | 7.7     | 0.0     | **5.8**  |
>    | **MTR-7B-Instruct**     | **18.0** | **15.4** | **8.3**  | **15.5** |
>
>    **Model on HumanEval benchmark**
>
>    | Model on HumanEval benchmark | HumanEval Pass@1 | Improvement        |
>    | ---------------------------- | ---------------- | ------------------ |
>    | Qwen2.5-7B-base              | 57.9%            | -                  |
>    | **MTR-7B**                   | **73.1%**        | **+15.2 points**   |
>
>    On GAIA, MTR improves average accuracy by 167% (5.8 → 15.5) with consistent gains across all levels (+12.9, +7.7, +8.3). Level 3 tasks exceed ToolMaker’s coverage, yet MTR still outperforms by 8.3 points, showing reasoning patterns generalize out-of-distribution. On HumanEval, MTR achieves 26% improvement on code generation despite no code tools in HotpotQA, demonstrating learned abstract tool orchestration rather than memorization.
>
> 3. We thank the reviewer for pointing this out and apologize for the lack of clarity regarding implementation details.
>
>    1. We now provide a detailed description of the data sources used in both the SFT and RL stages. In the SFT stage, we use a subset of the HotpotQA training split (100K questions in total) as our base dataset. For each question, we prompt the `claude-3-7-sonnet-20250219` model via the Anthropic API to generate high-quality reasoning traces. After filtering based on answer correctness (post-hoc verification against HotpotQA ground-truth answers), structural validity (no parsing errors and 2–12 tool calls), and the absence of degenerate behaviors (e.g., excessive looping), we retain approximately 60% of the traces (60K in total), as shown in line 233-238 of the main paper. These 60K traces are subsequently used for training in the SFT stage. In the RL stage, we sample an additional subset of 30K samples from HotpotQA and train GRPO using an online sampling strategy.
>    2. Prior to MTR training, we utilize the `claude-3-7-sonnet-20250219` model to generate high-quality reasoning traces. Specifically, the model is accessed via the Anthropic API and sequentially assumes the roles of *ToolMaker*, *AutoAgent*, and *ToolActor*; the only distinction among these roles is the system prompt provided (see Appendix A–C for details). For MTR, we instantiate *ToolMaker*, *AutoAgent*, and *ToolActor* using Qwen2.5-7B (either the base or instruct variants), with all three components sharing parameters during both training and inference. This means that a single Qwen2.5-7B model learns to perform all three roles jointly under shared weights. This setting is kept consistent across both the SFT and GRPO stages of MTR.

---

> ### Author Response · Authors · 2025-11-22
> **Response to Reviewer 1Ri8 Follow-Up (Part 2/2)**
>
> Then here are the answers to the Question part.
>
> 1. Please see the answers to Weakness 3.
> 2. Please see the answers to Weakness 3.
> 3. Please see the answers to Weakness 2.
> 4. We thank the reviewer for raising the question regarding simulated tool domains and generalization to unseen tools.
>
>    1. Our ToolActor generates diverse tool types across multi-hop QA tasks, including search tools (`google_search`, `scholar_search`, `wikipedia_search`), database tools (`demographics_search`, `biographical_database`), calculation tools (`calculator`, `statistics_analyzer`), and summarization tools (`answer_summarizer`). **See Appendix** for a full breakdown of 194K unique tools spanning these and other domains.
>    2. Our analysis demonstrates that MTR learns systematic and abstract tool-use patterns rather than benchmark-specific heuristics (see Appendix E.3 for more details). Specifically, ToolMaker generates 194K tools following a power-law distribution (R² = 0.985), with complexity adapted to task requirements. Moreover, as shown in Table 1 of the main paper, across four benchmarks, MTR achieves competitive results, confirming that the learned policy captures generalizable tool reasoning strategies. Notably, as discussed in our response to Weakness 2, in HumanEval—where code execution tools are entirely absent from the HotpotQA training data—MTR achieves 73.1% pass@1 (+15.2 points over the baseline), providing strong empirical evidence that the policy generalizes to unseen tool types.
>
> 5. We sincerely appreciate the reviewer's suggestion. We have carefully surveyed prior work on search-augmented reasoning and, to the best of our knowledge, SSRL (arXiv:2508.10874) listed in Table 1 is the most recent state-of-the-art baseline published before the submission deadline. SSRL proposes a self-search reinforcement learning approach where LLMs leverage their internal parametric knowledge through structured prompting, representing the latest advancement in this direction. If there are more advanced baselines that we may have missed, we would greatly appreciate it if the reviewer could point them out, and we would be happy to include experimental comparisons.
>
> 6. We thank the reviewer for raising this question. Our simulation explicitly considers **schema validation failures** as mentioned in line 175-177 of the main paper: when AutoAgent generates tool calls with type mismatches, range violations, or regex pattern errors, ToolActor returns structured error messages (e.g., `{"error": "Missing required parameter: query"}`), and AutoAgent must self-correct in subsequent steps. Ablation results (Table 5 in the main paper) confirm this capability: removing validation checks increases malformed calls from 2.1% to 9.2% and slot disagreements from 4.3% to 15.6%, demonstrating that the model learns effective self-correction. **Runtime failures** (e.g., timeouts, partial returns, network errors) are not explicitly simulated, by design. Our focus is on learning **strategic tool orchestration**—when and how to call tools and how to recover from logical errors—while deployment-specific API quirks can be addressed via post-training fine-tuning. Schema validation errors provide transferable reasoning skills, whereas runtime failures reflect infrastructure constraints rather than reasoning challenges.

---

### Official Review · Reviewer_PtcD · 2025-10-31

**Soundness:** 3
**Presentation:** 2
**Contribution:** 3
**Rating:** 6
**Confidence:** 3

**Summary:**

- Proposes MTR, a simulation‑first, multi‑agent framework for tool‑augmented reasoning that avoids live API calls by learning from complete ReAct traces with schema‑validated, simulated observations

- Architecture: ToolMaker (generates OpenAI‑compatible tool schemas), AutoAgent (think–act–observe), ToolActor (simulates tool responses)

- Two‑stage training: Stage‑1 SFT to learn “trace grammar”; Stage‑2 GRPO with a composite, trace‑level reward balancing answer correctness, internal consistency, and efficiency penalties

- On HotpotQA, MuSiQue, 2Wiki, Bamboogle, MTR matches live‑API systems on average EM (29.38% vs. 29.3%) and is notably better on Bamboogle (40.0% vs. 33.3%) despite no external APIs

**Strengths:**

- Removes API latency/cost brittleness while retaining competitive accuracy; especially strong on reasoning‑intensive evaluation

- Clear separation of structural vs. strategic competence (SFT→GRPO) with ablations showing each stage’s necessity

- Practical guardrails: JSON schema for tools and lightweight validation checks improve trace quality and training stability

- Sensible composite reward (final/intermediate consistency + loop penalty) and grouped sampling under GRPO

**Weaknesses:**

- Simulation–reality gap: ToolActor’s model‑generated observations may not capture real API noise, failures, or distribution shifts; deployment transfer remains untested beyond benchmarks

- Evaluation scope: Focused on multi‑hop QA; no results for broader tool ecosystems (e.g., program execution, structured DBs) or mixed simulated+live settings

- Trace curation bias: SFT relies on filtered “correct” traces (~60% retained), potentially narrowing exploration and overstating stability

- Complexity vs. gain: Multi‑agent tool generation/validation, trace filters, and GRPO add pipeline complexity without reporting wall‑clock, throughput, or cost per point EM; ROI vs. simpler single‑agent RAG/Agentic‑RL baselines is unclear

**Questions:**

See above.

---

> ### Author Response · Authors · 2025-11-22
> **Response to Reviewer PtcD Follow-Up (Part 1/2)**
>
> We appreciate your thorough review and recognition of the work's strengths, particularly the removal of API dependencies, clear separation of structural and strategic learning, and practical design choices.
> Here are the responses to Weakness part.
>
> 1. Thanks for raising this point. We acknowledge that simulation-first approaches entail an inherent trade-off between simulated and real tool interactions, but MTR’s design and results provide mitigating factors showing this does not significantly affect learning. First, MTR learns **abstract tool reasoning patterns** rather than memorizing API-specific behaviors: Table 5 in the main paper shows that removing validation checks increases malformed calls, and the tools-on vs. tools-off ablation (Table 4 in the main paper) shows substantial performance drops (e.g., Bamboogle: 40.0% → 20.7%) when tools are removed. Second, **simulation provides controlled, stable training**: ToolActor delivers consistent, schema-validated observations, and training stability analysis (Figure 5 in the main paper) shows significantly lower gradient variance compared to tool-free baselines. Third, **simulation quality is empirically validated**: ToolMaker generates 194K tools following a power-law distribution (R²=0.985) with task-adaptive complexity and diverse semantic coverage (Appendix A.6). Finally, this design tests a hypothesis about pre-trained models: MTR enables models to systematically access internal knowledge through structured tool reasoning. Competitive benchmark performance indicates that simulated observations effectively retrieve and structure knowledge encoded in model parameters, demonstrating that learned reasoning patterns are transferable and robust even without live tool execution.
>
> 2. We thank the reviewer for the suggestion to evaluate on a broader range of tasks. To demonstrate MTR’s wider applicability, we conducted experiments on code generation using the HumanEval benchmark (results shown in the table below). MTR achieves a 26% relative improvement over the base model. Notably, code execution tools (e.g., `python_execute`, `syntax_checker`) were entirely absent from the HotpotQA training data, yet MTR still attains substantial gains, demonstrating that the learned tool reasoning patterns transfer effectively to unseen domains and tool types.
>
> 	| Model           | HumanEval Pass@1 | Improvement      |
> 	|-----------------|------------------|------------------|
> 	| Qwen2.5-7B-base | 57.9%            | -                |
> 	| **MTR-7B**      | **73.1%**        | **+15.2 points** |

---

> ### Author Response · Authors · 2025-11-22
> **Response to Reviewer PtcD Follow-Up (Part 2/2)**
>
> 3. We thank the reviewer for the insightful comment. SFT aims to train the model to produce ReAct traces that are syntactically well-formed and semantically coherent, while exploratory capacity and reasoning diversity are addressed in the second-stage GRPO training. The 40% of filtered data primarily contained parse errors, validation failures, incorrect answers, or degenerate behaviors (e.g., excessive loops), which would negatively impact SFT learning. Therefore, using filtered, correct traces for SFT is appropriate.
>
> 4. We thank the reviewer for raising the question regarding efficiency and ROI. Here we report wall-clock (overall training time), training throughput, Cost per EM point and ROI vs. simpler baselines for a comprehensive efficiency analysis.
>
> 	1. **wall-clock**: we report the training cost in below table:
>
> 		| Stage | Dataset / Steps        | GPU               | training GPU-hours |
> 		|-------|------------------------|-------------------|--------------------|
> 		| SFT   | 60K traces, 2 epochs   | 8×H100 (80GB)     | ~36               |
> 		| GRPO  | 912 steps, rollout n=8 | 8×H100 (80GB)     | ~1202             |
> 		| Total | -                      | -                 | ~1238             |
>
> 	2. **Training throughput**: MTR achieves 707.5 tokens/second during GRPO training, processing 16.2M tokens over 912 steps. The simulation-first approach eliminates API latency bottlenecks, enabling consistent training throughput limited only by GPU compute.
>
> 	3. **Cost per point-EM**: Given that training GPU hours is 1238 and the average EM is 29.38, the Cost per EM point is 1,238 GPU-hours ÷ 29.38 = 42.1 GPU-hours/point.
>
> 	4. **ROI vs. simpler baselines**:
>
> 		| Method                 | Training Cost      | Cost per EM Point        | Avg EM (%)     |
> 		|------------------------|--------------------|---------------------------|----------------|
> 		| **Direct** (zero-shot) | None               | N/A                       | 14.5 (-14.88)  |
> 		| **CoT** (few-shot)     | None               | N/A                       | 17.4 (-11.98)  |
> 		| **RAG** (retrieval-only) | Minimal          | N/A                       | 18.8 (-10.58)  |
> 		| **MTR** (ours)         | **1,238 GPU-hours** | **42.1 GPU-hours/point** | **29.38**      |
>
> 		Given that the simpler baselines perform substantially worse than MTR, we believe that the performance gains afforded by this additional architectural complexity far outweigh the associated computational overhead.

---

### Note · Program_Chairs · 2026-01-17
**Submission Desk Rejected by Program Chairs**

The following references in this submission do not refer to real documents and/or have major errors in bibliographic information:

 Tianyu Zheng, Ge Zhang, Tianhao Shen, Xueling Liu, Bill Yuchen Lin, Jie Fu, Wenhu Chen, and Xiang Yue. Making tools for ai agents: A open source library for function generation and interface abstraction. arXiv preprint arXiv:2405.14011, 2024.